# The Role of Therapeutic Plasma Exchange (TPE) in Multisystem Inflammatory Syndrome in Children (MIS-C)

**DOI:** 10.3390/children8060498

**Published:** 2021-06-11

**Authors:** Gurkan Atay, Canan Hasbal, Mücahit Türk, Seher Erdoğan, Betül Sözeri

**Affiliations:** 1Department of Pediatric Critical Care, Health Science University, Ümraniye Research and Training Hospital, İstanbul 34760, Turkey; seher70@gmail.com; 2Department of Pediatrics, Health Science University, Ümraniye Training and Research Hospital, Istanbul 34760, Turkey; cananhasbal@yahoo.com (C.H.); mucahitturk_@hotmail.com (M.T.); 3Department of Pediatric Rheumatology, Health Science University, Ümraniye Training and Research Hospital, Istanbul 34760, Turkey; drbetulsozeri@gmail.com

**Keywords:** COVID-19, MIS-C, therapeutic plasma exchange

## Abstract

Multisystemic inflammatory syndrome in children (MIS-C) is a new potentially life-threatening disease that is related to coronavirus disease 2019 (COVID-19). The aim of this study is to reveal the clinical and laboratory results of MIS-C and the role of therapeutic plasma exchange (TPE) in its treatment. Clinical, laboratory and radiological characteristics of the patients who were admitted to the pediatric ward and pediatric intensive care unit (PICU) of a tertiary hospital with a diagnosis of MIS-C between April 2020 and March 2021 were included in the study. Forty-one patients were admitted to our hospital with a diagnosis of MIS-C. Twenty-one (51.2%) patients were admitted to the PICU. Six patients needed invasive mechanical ventilation (14.6%), 10 patients (24.4%) TPE and 3 patients (7.3%) needed extracorporeal membrane oxygenation (ECMO). The patients were grouped according to need for PICU admission (Group 1: no need for PICU, Group 2: need for PICU admission). Group 2 had significantly higher levels of C-reactive protein (CRP), alanine aminotransferase (ALT), ferritin, D-dimer, pro-B type natriuretic peptide (pro BNP) and lactate (*p* < 0.05). Hyponatremia found to be an independent risk factor for inpatient MIS-C in the PICU. We think that dynamic laboratory trending is beneficial in determining the need for PICU admission and TPE may be effective in critically ill patients.

## 1. Introduction

The COVID-19 pandemic, caused by severe acute respiratory syndrome coronavirus 2 (SARS-CoV-2), has caused nearly 100 million infections and more than 210,000 deaths worldwide as of January 2021 [1]. The disease tends to be milder in children, especially those without comorbid conditions, in comparison to adults [2,3,4]. However, a cluster of patients with an unexplained multisystem inflammatory syndrome with features reminiscent of Kawasaki disease, toxic shock syndrome, hemophagocytic lymphohistiocytosis and macrophage activation syndrome was reported in a group of pediatric intensive care units (PICUs) in April–May in UK [5]. Later, cases with a similar clinical picture were reported from the USA and other European countries [6,7].

Persistent fever, inflammation and organ failure in the absence of other infectious causes was dubbed “multisystem inflammatory syndrome in children” (MIS-C) by the Centers for Disease Control and Prevention (CDC) and the World Health Organization (WHO) and “pediatric inflammatory multisystem syndrome” (PIMS) by the Royal College of Paediatrics and Child Health (RCPCH), regardless of SARS-CoV-2 polymerase chain reaction (PCR) status [5,6,8]. The Centers for Disease Control and Prevention criteria, which include positive results for PCR, serology or antigen tests or exposure to a suspected or confirmed case of COVID-19 4 weeks prior to the onset of symptoms, were used for case definition.

In individuals under the age of 21, the estimated incidence of laboratory-confirmed SARS-CoV-2 infection was 322 per 100,000 and the incidence of MIS-C was only 2 per 100,000 [9].

Most patients with MIS-C have negative PCR but positive COVID-19 serology, which supports the notion that MIS-C is caused by immune dysregulation following SARS-CoV-2 infection. In reports of case series, 60% of the patients were reported to be PCR negative and serology positive, while 34% had both tests positive and 5% both tests negative [10,11,12,13]. Immunomodulation (IVIG, glucocorticoids, anakinra) and anticoagulant and/or antiaggregant agents are the mainstay of treatment in patients with MIS-C [14]. Information about the use and recommendation of extracorperal treatments in this disease is quite insufficient. Clinical and laboratory findings similar to macrophage activation syndrome have been observed in critical MIS-C cases [5].

Another study demonstrated that elevated inflammatory markers could indicate cytokine storm and macrophage activation syndrome [15]. Extracorporeal therapies like therapeutic plasma exchange (TPE) may have a role in treating and supporting these patients, especially in cases of severe cytokine storm, refractory vasoplegia and cardiogenic shock [16].

Therapeutic plasma exchange is effective in pediatric patients with macrophage activation syndrome [17]. There is also a recommendation for TPE in the American Society for Apheresis guidelines [18]. The aim of this study is to reveal the clinical and laboratory results of MIS-C and the role of TPE in its treatment.

## 2. Materials and Methods

Patients between the age of 1 month to 18 years, who were diagnosed with MIS-C according to CDC criteria, were included in the study. Clinical, laboratory and radiological characteristics of the patients who were admitted to the pediatric ward and PICU of a tertiary hospital with a diagnosis of MIS-C between May 2020 and March 2021 were evaluated.

Data collected from hospital records were used to fill out a standard form. The patients were grouped according to need for PICU admission. Group 1: no need for PICU admission, Group 2: need for PICU admission.

### 2.1. Data Collection

Age, gender, underlying disease, diagnosis on PICU admission, Pediatric Mortality Risk Score (PRISM) on PICU admission, duration of hospitalization before the PICU, antibiotic use, immunosuppressive use, mode and duration of mechanical ventilation, treatment with biologics, immunosuppressive treatment and extracorporeal treatment were recorded. Complete blood count, liver and kidney function tests, C-reactive protein (CRP), procalcitonin (PCT), ferritin, D-dimer levels and COVID-19 PCR and COVID-19 IgM and IgG results were recorded. Imaging findings (chest X-ray, computed tomography (CT) of the chest, ultrasound) were reviewed by a pediatric radiologist and recorded.

### 2.2. Statistical Analysis

Statistical analysis was performed using the Statistical Package for the Social Sciences (SPSS Inc; Chicago, IL, USA) 21. Normality was evaluated with Shapiro–Wilk tests and histograms. The data were expressed as median, minimum, maximum, frequency and percentage. Categorical variables were compared with the χ^2^ test or Fisher’s exact test when the expected sample size was <5. Normally distributed continuous variables were compared with the Student’s *t*-test. A Mann–Whitney U test was used for continuous variables that did not have a normal distribution. A *p* value of < 0.05 was considered statistically significant. In univariate analysis, significant predictors of inpatient and inpatient MIS-C in the PICU were used to perform a logistic regression analysis model to identify independent risk factors.

### 2.3. Ethics Committee and Informed Consent

This study was conducted with the approval of the Clinical Research Ethics Committee of Umraniye Research and Training Hospital (13.05.2020/9627). Informed consent was waived, as this was a prospective case–control study.

## 3. Results

Forty-one patients were admitted to our hospital with a diagnosis of MIS-C between March 2020 and March 2021. Twenty-one (51.2%) patients were admitted to the PICU. Median age was 10 (2–20), and 33 patients (80.5%) were male. Thirty-two patients (78%) had no underlying disease. Mean duration of fever was 5 days (3–8). Two patients (4.9%) were SARS-CoV-2 PCR positive and 37 patients (90%) had positive COVID-19 serology. All of these 37 patients were positive for COVID-19 IgG, while five (12.2%) patients were positive with a COVID-19 IgM test.

Gastrointestinal (GI) and cardiovascular findings were most common (19 patients) (46.3% each). Six patients needed invasive MV (14.6%), 10 patients (24.4%) needed TPE and 3 patients (7.3%) needed extracorporeal membrane oxygenation (ECMO). Patients who needed TPE and ECMO had elevated inflammatory markers, cytokine storm or cardiogenic shock which suggested macrophage activation syndrome.

Patients had 1 or 1.5 times their total plasma volume administered, using fresh frozen plasma in TPE. Patients had at least one and at the most 11 sessions, with a mean of five sessions of TPE performed. Median PRISM score of the patients that required PICU admission was 10. Two patients died during the study period (4.9%). Demographic and clinical characteristics of the patients are summarized in Table 1. The patients were grouped according to need for PICU admission (Group 1: no need for PICU admission, Group 2: need for PICU admission). All patients across both groups were given intravenous immunoglobulin (IVIG). Seventeen (41%) patients received vasoactive drugs and five patients (12.2%) received pulse methylprednisolone. Fifteen patients (36.6%) received the interleukin-1 receptor antagonist anakinra (two in Group 1, 13 in Group 2). Three patients in Group 2 received tocilizumab. Subcutaneous enoxaparin was given to 29 patients (70.7%), 10 patients were in Group 1 and 19 were in Group 2. Broad spectrum antimicrobials were empirically started in all patients in Group 2.

The two groups were similar in age, gender and duration of fever (*p* > 0.05). Group 2 had significantly lower lymphocyte count, thrombocyte count, sodium and albumin levels (*p*: 0.001, *p*: 0.011, *p*: 0.001 and *p*: 0.001, respectively). Group 2 had significantly higher levels of C-reactive protein (CRP), alanine aminotransferase (ALT), ferritin, D-dimer, pro-B type natriuretic peptide (pro-BNP) and lactate (*p*: 0.028, *p*: 0.028, *p*: 0.001, *p*: 0.028, *p*: 0.04 and *p*: 0.005, respectively). The two groups had similar hematocrit, leukocyte count and activated partial thromboplastin time (Table 2).

Five patients (12.2%) in Group 2 had patchy infiltrates and pleural effusion. Overall, the two groups were similar in imaging findings (*p*: 0.07). Twenty-three patients (56%) had systolic dysfunction and low ejection fraction on echocardiogram performed by a pediatric cardiologist. These findings were more common in Group 2 (*p*: 0.03). Five patients (14.6%) had dilatation of the coronary arteries.

A logistic regression analysis, including parameters with a *p* value < 0.05 found in the univariate analysis, was applied. Hyponatremia (*p*: 0.041, OR (95% CI): 1.8 (1.1–2.8)) was found to be an independent risk factor for inpatient MIS-C in the PICU.

## 4. Discussion

This study is important in terms of compiling the clinical, demographic and laboratory characteristics of MIS-C patients and revealing the risk factors that predict the need for PICU admission.

Studies have reported that 60% of MIS-C patients need inotropic drugs and/or fluid resuscitation, and 71% need PICU admission [13,19]. Feldstein et al. [12] reported that 62% of their MIS-C patients were male and the rate of PICU admission was 80%. In our study, 80.5% of the patients were male, and the rate of PICU admission was 51.2%. We thought that the lower rate of PICU admission compared to the literature was due to increased awareness of the disease because of the increase in the number of MIS-C patients in our hospital, the establishment of a diagnosis and treatment protocol for these patients and early diagnosis and rapid initiation of treatment.

Whittaker et al. [11], in their study evaluating 58 patients with MIS-C, stated that PCR was positive in 26% and serology (IgG) was positive in 87% of the patients. The most common clinical presentation was GI symptoms in that study, albeit at a lower rate than other studies (50%). We found a much lower rate of PCR positivity (4.9%) in our study, while serologic positivity was similar to the literature (90%). Most of our patients (46.3%) had prominent GI findings, and the occurrence of lymphadenopathy was similar (seven patients, 17.1%).

Kawasaki disease (KD), toxic shock syndrome, hemophagocytic lymphohistiocytosis and macrophage activation syndrome are four well-known hyperinflammatory conditions in children. They rarely require PICU admission. The characteristics that distinguish KD from MIS-C are that it shows a seasonal periodicity, patients are mostly under 5 years of age and the duration of fever is longer [20,21].

One study of clinical features suggests [20] that some patients (41%) with coronary dilatation may have had KD rather than MIS-C despite the abovementioned distinctive features. The low coronary dilatation rate (14.6%) in our study supports this idea that some patients with KD may have been diagnosed with MIS-C during the pandemic. The rate of conjunctivitis (26.8%), which is one of the important findings of KD, was lower in our study.

Li Jiang et al. [22], in their meta-analysis of patients with MIS-C, reported that the need for intensive care was 82%, the rate of vasoactive drug use was 62% and the need for invasive MV was 27%. Davies et al. [23] reported that 46% of 78 patients hospitalized in a PICU needed MV and 83% received inotropes. In our study, the need for vasopressors was similar, but the requirement for MV and respiratory support was lower. In the same study, it was reported that two of the three patients who needed ECMO died. Similarly, two of our three patients that needed ECMO died. Overall mortality in MIS-C is less than 2% in the literature [12,13,19]. The two patients we lost had comorbid illnesses and had presented in April 2020, when MIS-C had not been well defined and the treatment remained unclear. The fact that we did not lose MIS-C patients after that date supports this argument.

CRP, PCT, ferritin, IL-6, D-dimer and fibrinogen significantly increase while albumin decreases in patients with MIS-C. Leukocytosis and lymphopenia are detected while platelet counts may be low or normal [22,23,24]. Our laboratory data were in keeping with the literature. We found the sodium levels of PICU patients to be lower, with a median of 130 mmol/L, than in ward patients.

To the best of our knowledge, this is the first comparison of PICU and ward patients with MIS-C. We found lower sodium and albumin levels and lymphocyte counts and higher CRP, ALT, ferritin, D-dimer, pro-BNP and lactate levels in PICU patients. We think that these findings can guide clinicians in follow-up and in determining the need for PICU admission.

In studies of imaging in MIS-C, chest X-rays (CXR) were mostly normal, with infiltrations and pleural effusions in 15–25% of patients [22,24]. In our study, while the radiographs of the patients followed up in the child health clinic were normal, infiltrations and pleural effusions were observed in 23.5% of PICU patients, in keeping with previous reports.

Immunomodulation (IVIG, glucocorticoids, anakinra) and anticoagulant and/or antiaggregant agents are the mainstays of treatment [14]. Our local protocol was used to guide the management of MIS-C patients. Tocilizumab and TPE, although not yet included in treatment protocols, were used in 8.6% and 24.4% of our patients, respectively. Although TPE in adults with COVID-19 has been extensively studied, there are no pediatric studies as yet to our knowledge [25,26]. In studies, COVID-19 infection is shown to rapidly increase cytokines and chemokines as well as inflammatory cells such as neutrophils and monocytes that are attracted to the affected tissues, hence causing tissue damage as a result of excessive infiltration [27]. A large amount of proinflammatory cytokines (IL-6, IL1-B, TNF, etc.) are produced in mononuclear macrophages, and this results in a cytokine storm. Coagulopathy may also develop through direct viral damage or stimulation of platelet aggregation. The secretion of abnormal von Willebrand factor multimers (ULVWF multimers) lead to platelet-ULVWF accumulation in damaged endothelial cells and endotheliopaty-associated vascular microthrombotic disease. TPE treatment aims to reduce the abnormal procoagulant agents and cytokine load and replace the missing coagulation factors.

Adult studies state that TPE provides clinical benefit by replacing missing plasma factors, removing inflammatory mediators and strengthening immune regulation and reticuloendothelial system functions. Khamis et al. [28] performed TPE on 11 of 31 adult patients with COVID-19 and reported that the patients who underwent TPE had a higher extubation rate (*p*: 0.018) and a lower 28-day mortality (*p*: 0.033). Additionally, we know that macrophage activation syndrome clinic and laboratory findings have been demonstrated in MIS-C cases [5]. A study by Demirkol et al. [29] showed that in 23 cases with MAS, TPE was effective in lowering ferritin values, increasing platelet values, and lowering lactate dehydrogenase levels from the first session. TPE was planned for patients who had a poor clinical condition and low platelet values and high ferritin values despite IVIG and steroid treatment. In eight of the patients, TPE treatment was successful (fall in ferritin values, increase in platelet count, regression of clinical findings), while two patients had partial regression of laboratory results but no resolution of clinical findings. We see that those patients had chronic diseases and died without benefiting from ECMO treatment.

## 5. Conclusions

As SARS-CoV-2 continues to wreak havoc globally, up-to-date information on MIS-C is crucial to guide management. Early PICU admission can be lifesaving. In light of our findings, we think that dynamic laboratory trending is beneficial in determining the need for PICU admission and TPE may be effective in critically ill patients.

## Figures and Tables

**Table 1 children-08-00498-t001:** Characteristics of Patients.

Total patient number	41
Age, years, median (interval)	10(2–20)
Gender, male, percentage (%)	33(80.5)
Weight, kilogram, median(interval)	32(9–109)
PICU admission, percentage (%)	21(51,2)
Duration of fever, day, median (interval)	5(3–8)
Concomitant chronic disease, percentage (%)	8(19,5)
Regular drug usage, percentage (%)	6(14,63)
History of covid positive case contact, percentage (%)	18(43,9)
Covid PCR positivity, percentage (%)	2(%4,9)
Covid Antibody positivity, percentage (%)	37(%90,2)
**Clinical presentation**	
▪ Upper Respiratory Tract Findings, percentage (%)	11(26,8)
▪ Lover Respiratory Tract Findings, percentage (%)	9(21,9)
▪ Hematological Findings, percentage (%)	1(2,4)
▪ Central Nervous System Findings, percentage (%)	5(12,2)
▪ Gastrointestinal System Findings, percentage (%)	19(46,3)
▪ Cardiovascular System Findings, percentage (%)	19(46,3)
▪ Urinary System Findings, percentage (%)	4(9,7)
▪ Artralgia Findings, percentage (%)	6(14,6)
▪ Presence of lymphadenopathy, percentage (%)	7(17,1)
▪ Presence of conjuktivitis, percentage (%)	11(26,8)
**PICU * Number of Inpatients**	21
Duration of hospitalization in PICU, day, median (range)	5(1–75)
Total duration of hospitalization, day, median (range)	9(4–92)
PRISM * score at the administiration of PICU, median (range)	12(5–37)
PELOD * score at the administiration of PICU, median (range)	10(0–52)
Number of patients on MV *, percentage (%)	5(14,3)
**Number of patients applied TPE *, percentage (%)**	**10(24,4)**
Number of patients applied ECMO *, percentage (%)	3(8,6)
Mortalite, percentage (%)	2(5,7)

* PICU: Pediatric Intensive Care Unit, ECMO; Ekstra Corporoel; Membrane Oxygenation, TPE; Therapeutic Plazsma Exchange; MV; Mechanical Ventilation, PRISM; Pediatric Mortality Risk Scoring. PELOD; Pediatric Logistic Organ Dysfunction Score.

**Table 2 children-08-00498-t002:** Comparison of the clinical, laboratory and radiological findings of the patients with those who were hospitalized in the PICU and non- PICU.

	Hospitalized in the PICU	Not Hospitalized in the PICU	*p*
Age [mounth, median (range)]	12(5–17)	8(2–20)	0.17
Duration of fever [day, median (range)	5(3–8)	5(3–8)	0.629
Duration of hospitalization [day, median (range)]	15(7–92)	9(4–27)	0.004
Gender [male, percentage]	15(71,4)	18(90)	0.33
Laboratory parameters, median (range)			
Hematocrit (%)	32(27–42)	35.5(28–44)	0.388
White blood cell count (cell/µL)	8480(3680–31,340)	10,880(1950–21,970)	0.234
Absulute lymphocyte count (cell/µL)	480(350–1200)	1590(440–6320)	0.001
Trombocyte count (cell/µL)	195,000(69,000–423,000)	314,500(97,000–715,000)	0.011
C-reactive protein (mg/L)	155(13–290)	40(3–270)	0.028
Eritrocyte sedimentation rate (mm/hr)	36,5(14–92)	45(7–63)	0.695
Fibrinogen(g/L)	607(113–907)	525(290–859)	0.869
Creatinin (mg/dL)	1(0.5–1.2)	1(0.4–1.1)	0.952
Sodium(mmol/L)	130(124–143)	136(131–145)	0.001
AST (U/L)	33(14–162)	29(14–69)	0.621
ALT (U/L)	30(7–92)	19(9–68)	0.028
GGT (U/L)	51(6–248)	18(7–45)	0.129
LDH (U/L)	323(172–1800)	283(174–545)	0.4
Ferritin (ng/ML)	1200(367–22,627)	332.5(16–3274)	0.001
Triglyceride (mg/dL)	180(96–383)	121.5(41–239)	0.129
Albumin (g/dL)	3(2–4)	4(3–5)	0.001
PT	17(12–25)	16(13–20)	0.316
aPTT	32(28–42)	29(26–40)	0.952
D-dimer (mg/L)	4400(1293–8400)	1735(200–14,100)	0.028
Pro-BNP	790(165–7145)	154(12–1135)	0.04
Lactate	3(1–21)	1(1–4)	0.005
Radiological Findings			
infiltration & pleural effusion in X-ray (%)	4(23,5)	0	0.07
Presence of the ecocardiography finding, (%)	13(76,5)	6(33,3)	0.03
intraabdominal fluid in a ultrasoun (%)	6(35,3)	8(44,4)	0.72

PICU: Pediatric Intensive Care Unit.

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
