# Peer review of "The Role of Therapeutic Plasma Exchange (TPE) in Multisystem Inflammatory Syndrome in Children (MIS-C)"

_children, 2021, doi:10.3390/children8060498_

Round 1

Reviewer 1 Report

The authors sought to describe the clinical and laboratory results of MIS-C and to describe the role of therapeutic plasma exchange (TPE). The following are my comments and suggestions for future revisions:

Abstract: 

  • I would include the full name of TPE when first described (second sentence), rather than later in the abstract. 
  • I would spell out mechanical ventilation rather than use the abbreviation MV. 
  • Based on the abstract, the grouping of the patients is unclear but is further described in the body of the manuscript. I would suggest clarifying the abstract. 
  • I don’t think all of the individual p-values are necessary in the abstract, unless requested specifically by the journal. 
  • Based on the abstract, the results suggest that the primary aim of the study was to identify risk factors for PICU admission or at least to identify differences among those admitted to the PICU. If so, I would recommend adjusting the aims of the study and title of the manuscript accordingly. 

Introduction: 

  • The definitions of MIS-C and PIMS are slightly different, but the statement on page 2 (line 47) suggests that all the definitions do not account for the SARS-CoV-2 test status. I would clarify, as the CDC definition requires exposure or evidence of current or (more often) past infection. 
  • I presume there is an error in the sentence on page 2 (lines 59-60) about macrophage activation syndrome. 
  • Has TPE been reported elsewhere in MIS-C? Why would it be helpful? It would be helpful for the authors to explain the rationale further for MIS-C.

Results: 

  • The authors state that 10 patients needed TPE. What were the indications for TPE in these MIS-C patients? That has not been the typical therapy for MIS-C, but of course, if it's beneficial in certain cases, that will be helpful for the medical community. 
  • At times, the results were unclear – specifically clarifying which cohort (no PICU or PICU) to which the labs belonged. The table is helpful though, and I would suggest revising the results section accordingly.

Discussion: 

  • I believe there is an error in this sentence to indicate “no studies,” and the 27th reference is included twice: “Although TPE in adults with COVID-19 has been extensively studied, there are pediatric studies as yet to our best knowledge (27, 27).” 
  • While reading the study, I was curious about the decision to use TPE. It was not until the last paragraph of the discussion though that the authors provided an explanation and background. This information would be helpful earlier in the study. 

Overall, I commend the authors for their treatment of these patients during this pandemic and for writing this manuscript. There have been other studies though that have addressed risk factors for severe disease and ICU admission, per my recollection. While that information is helpful, this study would be unique in regards to TPE. I would suggest focusing more on TPE – i.e. why your institution decided to use TPE in these patients, expanding the literature review of TPE, etc.

Reviewer 2 Report

The authors made a cross sectional observational study titled: The role of therapeutic plasma exchange therapy (TPE) in Multisystem inflammatory syndrome in children (MIS C).

Few remarks:

  1. Abbreviations are in abstract without previous description (MV)
  2. TPE abbreviation is explained in abstract after it has already been used in several sentences prior to description (line 18- line 23), should be, at least, inversed.
  3. Materials and methods: Authors should state the proper type of study. Authors should clearly state of inclusion and exclusion criteria.
  4. Line 91- please provide IRB reference number.
  5. Line 102 and 152 – I am not introduced with “hasta”, can you please explain.
  6. Line 194/195 – please revise sentence
  7. Text should be edited – spaces, letter capitalization, missing letters, P value forms)

Round 2

Reviewer 1 Report

The authors have addressed all of the reviewers' comments, to my knowledge, and have provided further clarification of the TPE role in MIS-C. I think the manuscript is stronger and provides helpful information.